# Impact of Phenylketonuria on the Serum Metabolome and Plasma Lipidome: A Study in Early-Treated Patients

**DOI:** 10.3390/metabo14090479

**Published:** 2024-08-30

**Authors:** Jorine C. van der Weerd, Annemiek M. J. van Wegberg, Theo S. Boer, Udo F. H. Engelke, Karlien L. M. Coene, Ron A. Wevers, Stephan J. L. Bakker, Pim de Blaauw, Joost Groen, Francjan J. van Spronsen, M. Rebecca Heiner-Fokkema

**Affiliations:** 1Department of Laboratory Medicine, Laboratory of Metabolic Disease, University Medical Center Groningen, University of Groningen, 9700 RB Groningen, The Netherlandsj.groen@umcg.nl (J.G.); 2Division of Metabolic Diseases, Beatrix Children’s Hospital, University Medical Center Groningen, University of Groningen, 9700 RB Groningen, The Netherlands; 3Department of Human Genetics, Translational Metabolic Laboratory (TML), Radboud University Medical Center, 6525 GA Nijmegen, The Netherlands; 4Laboratory of Clinical Chemistry and Hematology, Máxima Medical Centre, 5504 DB Veldhoven, The Netherlands; 5Division of Nephrology, Department of Internal Medicine, University Medical Center Groningen, University of Groningen, 9700 RB Groningen, The Netherlands; s.j.l.bakker@umcg.nl

**Keywords:** untargeted, metabolomics, lipidomics, phenylketonuria, mass-spectrometry, phenylalanine

## Abstract

Background: Data suggest that metabolites, other than blood phenylalanine (Phe), better and independently predict clinical outcomes in patients with phenylketonuria (PKU). Methods: To find new biomarkers, we compared the results of untargeted lipidomics and metabolomics in treated adult PKU patients to those of matched controls. Samples (lipidomics in EDTA-plasma (22 PKU and 22 controls) and metabolomics in serum (35 PKU and 20 controls)) were analyzed using ultra-high-performance liquid chromatography and high-resolution mass spectrometry. Data were subjected to multivariate (PCA, OPLS-DA) and univariate (Mann–Whitney U test, *p* < 0.05) analyses. Results: Levels of 33 (of 20,443) lipid features and 56 (of 5885) metabolite features differed statistically between PKU patients and controls. For lipidomics, findings include higher glycerolipids, glycerophospholipids, and sphingolipids species. Significantly lower values were found for sterols and glycerophospholipids species. Seven features had unknown identities. Total triglyceride content was higher. Higher Phe and Phe catabolites, tryptophan derivatives, pantothenic acid, and dipeptides were observed for metabolomics. Ornithine levels were lower. Twenty-six metabolite features were not annotated. Conclusions: This study provides insight into the metabolic phenotype of PKU patients. Additional studies are required to establish whether the observed changes result from PKU itself, diet, and/or an unknown reason.

## 1. Introduction

Phenylketonuria (PKU, McKusick/OMIM# 261600) is a rare autosomal recessive disorder affecting aromatic amino acid metabolism. Worldwide, PKU has an estimated incidence of 1 in 23,930 live births [1]. The disease is characterized by mutations in the gene encoding phenylalanine hydroxylase (PAH; EC 1.14.16.1; OMIM* 612349), an enzyme responsible for converting L-phenylalanine (Phe) into L-tyrosine (Tyr). In individuals with PKU, the activity of the enzyme PAH is significantly reduced or absent, leading to hyperphenylalaninemia (HPA), which is the accumulation of Phe in blood and brain tissue. HPA is not specific for PKU because it may also be a consequence of other hereditary causes, like defects in the pterin metabolism, resulting in a deficiency of the PAH cosubstrate tetrahydrobiopterin (BH4) and DNAJC12 [2,3]. Depending on the residual enzyme activity, patients have a widely varying Phe tolerance. If late-treated, PKU can lead to progressive and irreversible damage to the brain, resulting in intellectual disability [4,5,6,7,8,9,10,11,12].

Systematic measurement of Phe in dried blood spots or plasma is crucial to the diagnosis and monitoring of treatment of PKU patients [13]. Management of PKU primarily consists of a natural protein-restricted diet enriched with minerals, vitamins, and other micronutrients and/or low Phe casein glycomacropeptide to keep blood Phe concentrations within an acceptable range [2,13,14,15]. Supplementation is essential for meeting nutritional needs, supporting normal growth, and preventing developmental deficiencies [2,13,14,15]. A subset of patients may benefit from sapropterin dihydrochloride, an orally active synthetic form of tetrahydrobiopterin that enhances the residual activity of PAH. All patients may benefit from pegvaliase, which converts Phe into harmless compounds that can be excreted from the body. Both drugs lower Phe concentrations and enable patients to relax their protein restriction. However, pegvaliase is only reimbursed in a minority of countries around the world [16,17,18]. Except perhaps for treatment with large neutral amino acids (LNAAs), intensive Phe monitoring is crucial but is not able to predict the clinical outcome fully [19]. For this reason, there is a need to expand the biomarker signature beyond blood Phe to better reflect neurocognitive and psychosocial function.

Over the years, high-resolution mass spectrometry (HRMS) has become a powerful tool for discovering novel disease biomarkers for diagnosis and monitoring various inborn errors of metabolism, including PKU [20,21]. This technique features high resolution and, thus, high mass accuracy, enabling comprehensive profiling of known and unknown metabolites in complex biological matrices. In the last decade, the impact of PAH deficiency on the metabolome has become evident [21,22]. The levels of Tyr and downstream metabolites are lower, while the alternative Phe degradation pathway results in higher levels of phenylpyruvate and other (neuro)toxic metabolic products. In contrast, the non-polar part of the metabolome remains rather unexplored. Guerra et al. [23] emphasized the need to study the lipidome in this group of patients. Therefore, we compared the plasma lipid and serum metabolite signature of adult PKU patients with non-PKU controls.

## 2. Materials and Methods

Chemicals. The following chemicals were used for lipidomics analyses: methyl tert-butyl ether UPLC-grade (MTBE; Biosolve BV, Valkenswaard, The Netherlands), methanol absolute UPLC-grade (MeOH; Biosolve BV, Valkenswaard, The Netherlands), Milli-Q (MQ; Millipore, MA, USA), chloroform (CHCl_3_; Merck KGaA, Darmstadt, Germany), acetonitrile (ACN; Biosolve BV, Valkenswaard, The Netherlands), 2-propanol (IPA; Biosolve BV, Valkenswaard, The Netherlands), and ammonium formate UPLC-grade (AmF; Sigma-Aldrich, Zwijndrecht, The Netherlands). The following non-physiologic or stable isotope-labeled lipid standards were purchased from Avanti Polar Lipids, Inc (Alabaster, AL, USA): LPC(17:1/0:0), PC(17:0/17:0), LPE(17:1/0:0), PE(17:0/17:0), LPG(17:0/0:0), PG(17:0/17:0), LPS(17:0/0:0), PS(17:0/17:0), CL(14:0.14:0)(14:0/14:0), CL(16:0/16:0)/(16:0/16:0), DG(16:0/16:0/0:0), and TG(17:0/17:1/17:0)d5. All lipids were dissolved in CHCl_3_/MeOH/MQ (60:30:4.5, *v*/*v*/*v*) or chloroform to a final concentration of 20 µM and stored at −20 °C (compound details are provided in Appendix A). Chemicals used for metabolomics analyses are described elsewhere [21].

PKU samples. The PKU-COBESO study, a multicenter research project conducted in The Netherlands, focused on exploring the cognitive, behavioral, and social outcomes in relation to metabolic control in early-diagnosed and treated PKU patients [24,25,26,27]. The study included children and adults (age [7–42 years]). Participants were included between 2012 and 2015. Venous blood samples were taken after an overnight fast on the day of testing before the neuropsychological assessment. The EDTA- and serum samples were stored for up to 11 years at −80 °C without thawing before analysis. In the current study, individuals under the age of 18 were excluded from participation.

Control samples. No blood samples were taken from the control subjects in the COBESO study. For this reason, blood samples of the control group for the current study were sourced from pre-transplant kidney donors through the TransplantLines Biobank [28]. Transplantlines is a longitudinal cohort study investigating the outcomes after transplantation in patients in the UMCG. We consider this control group to be a good representative of the general population, with the sole selection criteria as kidney donors that they have two kidneys and a normal kidney function. We opted to use control samples from this cohort study rather than obtaining new samples from healthy individuals to partially account for any potential impacts of extended frozen storage. The EDTA- and serum samples from the TransplantLines Biobank, which were gathered between 2016 and 2020, have been preserved at −80 °C before their analysis.

For the statistical analysis, samples were matched by age and gender. In lipidomics, 22 PKU samples (median age: 33, age range: 21–43, gender ratio: 14M/8F) were compared with 22 control samples (median age: 34.5, age range: 24-43, gender ratio: 14M/8F). In metabolomics, 20 PKU serum samples (median age: 33, age range: 21–43, gender ratio: 12M/8F) were compared with 20 control samples (median age: 34.5, age range: 24–43, gender ratio: 12M/8F). In total, 15 metabolomics samples could not be matched with a suitable control; these were measured but were not included in the statistical analysis.

### Lipidomics LC-MS Workflow and Data Processing

Lipid extraction. Lipid extraction was performed with the one-phase MMC extraction method, as previously described by Gil et al. [29], with slight modifications. In short, a volume of 40 μL of an internal standard mix containing 12 non-physiologic or stable isotope-labeled lipid standards was added to an Eppendorf tube and vacuum dried at 40 °C. Afterward, 10 μL of EDTA-anticoagulated plasma and 200 μL of MMC extraction solution containing MeOH/MTBE/CHCl_3_ (1.33:1:1, *v*/*v*/*v*) was added. The samples were vortexed (~10 s) and incubated on an Eppendorf Thermomixer (Eppendorf, Hamburg, Germany) at a rate of 800 rpm for 1 h at room temperature. Next, the samples were vortexed again and centrifuged at a speed of 1000 g for 10 min at 4 °C. Subsequently, the supernatant was collected and transferred into a clean Eppendorf tube with a spin filter. The extraction procedure was repeated with 40 μL MMC extraction solution, and the supernatant was collected and combined with the first fraction. The samples were centrifugated for 5 min at 1020 RCF. All samples were vacuum dried at 40 °C. In the final step, the lipids were dissolved in a final volume of 100 μL composed of 25 μL CHCl_3_/MeOH/MQ (60:30:4.5, *v*/*v*/*v*) and 75 μL ACN/IPA/MQ (2:1:1, *v*/*v*/*v*). All samples were transferred into a vial with an insert shortly before LC-MS analysis.

Lipidomic analysis. Plasma total triglyceride (TTG) and cholesterol (TC) concentrations were analyzed using the Roche Triglycerides (08058687190) and Cholesterol Gen.2 (08057443190) kits on a Cobas Pro analyzer (Roche Diagnostics, Risch-Rotkreuz, Switzerland). In 2021, untargeted lipidomic analysis was performed using ultra-high-performance liquid chromatography-electrospray ionization mass spectrometry (UPLC-ESI MS) on a Synapt G2-Si high-resolution quadrupole time-of-flight (QTOF) mass spectrometer equipped with a Jetstream ESI source (Waters, Manchester, UK), as described by Gil et al. [29], with some minor adjustments. Liquid chromatography was performed on an Acquity UPLC CSH C18 column (2.1 × 100 mm 1.7 μm) (Waters, Manchester, UK) column, using a binary solvent system with mobile phase A (10 mM AmF in Milli-Q) and B (10 mM AmF in MeOH). The column was pre-heated to 80 °C, and the autosampler was set to 15 °C. Lipid extracts (2 μL) were injected and separated with a flow rate of 0.5 mL/min. Linear gradient elution proceeded as follows: 0 min 40% eluent A, 7.5–15 min from 40 to 10% eluent A, followed by an isocratic elution of 0% eluent A and 100% eluent B for 17.6 min.

The QTOF settings were as follows: the capillary voltage was 0.50 kV for ESI(+) and 0.70 kV for ESI(−); the cone voltage of the samples was set at 30 V; sampling cone 20 °C, the desolvation gas temperature was set at 600 °C, the desolvation gas flow was set at 1000 L/h, source offset 20, source temperature 120 °C, cone gas 100 L/h and nebulizer gas 6.5 bar. Accurate mass determination was ensured using 0.1 ng mL^−1^ leucine-enkephalin solution ([M + H]^+^ = 556.2771 and [M − H]^−^ = 554.2615) as the lock mass solution. The instrument was operated in data-independent acquisition mode. Fragmentation was achieved with argon gas at a low energy of 1 kV and a high energy of 30–60 kV. Data were collected over the m/z range from 50 to 2000 Da in high-resolution mode for ESI(+) and resolution mode for ESI(−) with an acquisition rate of 1 spectrum/0.3 s. Detailed information about UPLC and MS parameters is provided in Appendix A.

Batch design and quality control. To reduce and monitor analytical variation and bias, each batch included (i) randomization of the sequence order and (ii) stabilization of the UHPLC-QTOF before the actual sequence of samples was measured by injecting 10 times the quality control (QC) sample before each batch, (iii) measuring a QC sample at regular intervals (every 10–15 samples) in both series to evaluate the stability and repeatability of the system, (iv) injection of blank samples with and without IS, and (v) the use and analysis of internal non-physiological and/or isotopically labeled standards in the QC samples (see also Appendix A). All samples were measured in singlicate.

Metabolomic analysis. Serum samples from PKU patients and controls were analyzed by the UHPLC-ESI-QTOF-MS method previously described in detail for heparin-anticoagulated plasma, with some minor adjustments [21]. All serum samples were measured in duplicate and antiparallel order. The measurements were conducted in 2021 for ESI(+) and in 2023 for ESI(−), respectively.

Data processing. Both lipidomics and metabolomics data are available upon reasonable request but only in anonymized form. Both datasets were processed in the same manner. A schematic description of the data processing workflow is shown in Figure 1. Waters or Agilent raw data files were analyzed using Progenesis QI software Version 2.4 (Nonlinear Dynamics, Newcastle, UK) for alignment, peak picking, and deconvolution. Data were aligned on a reference run that was automatically selected from all QC measurements before peak picking. The run used as a reference file was most representative of all QC runs within or between batch(es). Alignment was considered to be successful when the score was above 95%. Runs with a score below the threshold were excluded. Afterward, the data were subjected to automatic peak picking. The following list of adducts was used for peak picking in ESI(+) mode: [M + H]^+^, [M + NH_4_]^+^, [M + Na]^+^, [M + K]^+^, [M + 2Na − H]^+^, [M+H − H_2_O]^+^, [M + H-2H_2_O]^+^, [M + CH_3_OH + H]^+^, [2M + H]^+^, [2M + NH_4_]^+^, [2M + Na]+, [2M + K]^+^, [M + 2H]^2+^, [M + H + Na]^2+^, [M + 2Na]^2+^, [M + 2H + Na]^3+^, [M + 2Na + H]^3+^. The following adducts were used in ESI(−) mode: [M − H]^−^, [M − H_2_O − H]^−^, [M + Na − 2H]^−^, [M + K − 2H]^−^, [M + FA − H]^−^, [2M − H]^−^, and [2M + FA − H]^−^.

Peak picking limits were set at ‘default’ sensitivity mode. Features were excluded if the predominant adduct detected had a charge (z) of 2 or 3 (i) and/or when the isotopic pattern was absent (Appendix A) (ii). Subsequently, the detected compound ions were deconvoluted to generate the final list with features. Raw feature abundances were exported from Progenesis QI. Data processing was performed with MetaboAnalystR [30]. Features with 80% missing values were excluded and missing values were imputed with 1/5 of the minimum positive value of each variable. Data were filtered on the pooled quality control samples that were included in each run (coefficient of variation (CV) < 30%).

Multivariate analyses. Multivariate analyses (MVA) were performed with SIMCA P version 17.0 (Umetrics AB, Umea, Sweden) by using principal component analysis (PCA) and orthogonal projections to latent structures (OPLS) analysis (PKU versus Control). Metabolomics data were normalized on L-phenyl-D5-alanine before MVA. Lipidomics data were normalized on the lipid standard that was most representative of the feature (e.g., LPC(18:0) was normalized on the standard LPC(17:1)); this was possible only after annotation following MVA. If there was not a suitable standard, the average intensity of the standards detected in the sample was taken. Before analysis, the lipidomics and metabolomics datasets were log-transformed and Pareto-scaled. For each MVA model, the quality was controlled by computing Hotelling’s T2 statistics. Model performance is reported as cumulative correlation coefficients for the model (R2), predictive performance based on K-fold cross-validation (Q2), as well as cross-validated ANOVA (CV-ANOVA) *p*-values for OPLS-based group separation. Feature selection was performed using variable importance in projection (VIP) scores, applying a cutoff for the VIP of greater than 0.5.

Compound reviewing. To gain confidence in the validity of the discriminating compounds from MVA, each feature was extensively reviewed. From the list of discriminating features, we aimed to determine the number of unique features. The number of features detected does not necessarily reflect the number of unique features present in the dataset [31]. Therefore, we aimed to remove redundant features before proceeding to feature annotation. Detailed information about compound reviewing is reported in Appendix A. Reasons to reject features for further analysis were: (A) Deconvolution failure: compound ions are grouped by the algorithm but do not belong to the same feature; (B) Noise: features that were considered noise based on manual inspection of the peak shape of the raw extracted ion chromatograms; (C) Chromatographic resolution failure: features that were integrated with other compounds that were not separated chromatographically or not recognized by the algorithm as different features; (D) In-source fragmentation (ISF): features that were suspected of being a result of ISF. This type of feature was recognized based on the identical abundance pattern, peak shape, and retention time.

Univariate analysis. Univariate statistical analysis was employed to evaluate whether the age difference, total triglyceride content (TTG), total cholesterol (TC), and normalized feature abundances were significant. For this, the Wilcoxon rank sum test (*p* < 0.05; false discovery rate (FDR) adjusted, Benjamini–Hochberg) was applied. We used the Benjamini–Hochberg procedure to control the false discovery rate, i.e., the proportion of “discoveries” (significant results) that are actually false positives [32]. Chi-squared test was conducted to evaluate whether the sex difference was significant (*p* < 0.05) between matched PKU and control samples.

Feature annotation and nomenclature. Features were putatively annotated with mass information present in public databases. In both lipidomics and metabolomics data, the accurate mass could not deviate more than |5| parts per million (ppm) from the theoretical mass. The lipidomics features were annotated based on information from the LipidMaps structure database and ChemSpider databases [33,34]. The mass spectra were checked for the presence of qualitative fragments that were previously described in 2015 by T’Kindt et al. [35] using a comparable MS method (Appendix A). All lipids described in this article were annotated according to the nomenclature proposed by the International Lipid Classification and Nomenclature Committee [36,37]. The majority of the lipid features were annotated at the subclass level, e.g., PG(34:1). Only lysophospholipids annotations were made at the fatty acyl level, e.g., PG(16:0_0:0).

For metabolomic features, the Human Metabolome Database (HMDB) was used in combination with an in-house IEM panel that consisted of 340 IEM-related metabolites (version 0.8) [38,39,40,41,42]. Both lipidomic and metabolomic feature annotations were systematically classified by the guidelines pointed out by the Metabolomics Standards Initiative [43]. Features were annotated across four distinct levels: identified metabolites or lipids (level 1; confirmed by an authentic chemical standard analyzed under the same conditions, matching at least two orthogonal criteria here, accurate mass and retention time), putatively annotated features (level 2), putatively characterized feature classes (level 3), and features of unknown identity (level 4). In the last case, the identity was reported as the m/z value, or if possible, the neutral mass calculated by the software after deconvolution of the compound ions.

Correlation with concurrent dried blood spot Phe levels. Spearman’s rank-order correlation analysis was performed to explore the relationship between concurrent dried blood spot Phe levels and the statistically different features. All results are given as two-tailed *p*-values, and *p* < 0.05 was considered statistically significant.

Pathway enrichment analysis. Over-representation analysis (ORA) was performed using the pathway analysis module from the web-based platform MetaboAnalyst 6.0. Annotated metabolites with a VIP > 0.5 were included. Metabolites whose identification was uncertain (e.g., due to isomerism) or that lacked an available KEGG compound identification number were excluded from the analysis. The analysis was facilitated using the hypergeometric test. Human pathways were collected from the KEGG (KEGG pathway information was obtained in December 2023). To assess the significance and control for false discoveries in the pathway analysis, both the raw *p*-value and the false discovery rate (FDR) *p*-value were considered. The established thresholds for statistical significance were set at raw *p*-value < 0.05 and FDR *p*-value < 0.1.

## 3. Results

### 3.1. Population Characteristics

For all PKU patients, compliance to diet was controlled by regular monitoring of blood Phe levels. Therefore, for the majority of the patients, concurrent Phe levels were available. All patients from this study were diagnosed early by neonatal screening and treated continuously at clinical centers in The Netherlands. Treatment consisted of diet, sapropterin therapy, or both. All characteristics are displayed in Table 1 and Appendix A. None of the selected controls was diagnosed with HPA or PKU, and Phe levels were, therefore, considered to be normal (local reference range for adults: 35–85 µmol/L). For the (sub)sets used for MVA of the metabolomics and lipidomics datasets, the age and sex of PKU and controls did not differ significantly.

### 3.2. Data Processing

For lipidomic analysis, all samples were measured in a single batch in both ESI(+) and ESI(−) modes. Two samples for which alignment failed were excluded from the analysis. Processing of the raw *lipidomics* data in Progenesis QI resulted in the detection of 15,220 and 5223 features in ESI(+) and ESI(−) ion mode, respectively. After the exclusion of features without an isotopic pattern, with a charge of 2+ or 3+, and with a high (>30%) CV in the pooled sample (Table 2), 30% (4438 features, ESI(+)) and 84% (4416 features, ESI(−)) of the features met our criteria. Metabolomics was performed in two separate batches, ESI(+) and ESI(−) modes, resulting in four datasets in total. The first batch contained 20 control samples and 20 PKU samples, the second batch included also the 20 control samples and the remaining 15 PKU samples. Processing of the raw *metabolomics* data in Progenesis QI resulted in the detection of 3627 and 2258 features in ESI(+) and ESI(−) ion modes, respectively. Both batches were processed in a single Progenesis QI experiment; as a result, the outcomes of some processing steps (e.g., peak picking) are consistent for batches A and B. After applying the primary exclusion criteria (Table 2), 20% (725 features, ESI(+)) and 22% (503 features, ESI(−)) of the features passed the criteria in both batches and were subjected to further statistical analyses.

### 3.3. Lipidomics

#### 3.3.1. Multivariate Analyses of Plasma Lipidomic Data

Total TG content was statistically significantly higher in PKU patients compared to controls (*p* = 0.04), while there was no difference in plasma total cholesterol concentrations. Unsupervised PCA was applied to the *lipidomic* datasets as an exploratory data analysis tool to gain insight into quality, general trends, and intrinsic clustering of samples (Appendix A). The PCA score plot revealed clustering according to the experimental group, indicating different lipid phenotypes for PKU and controls (Figure 2).

To assess which features are most discriminating between both groups, supervised MVA was employed. For this, an OPLS-DA model was constructed including the features that remained after filtering. For both ESI(−) and ESI(+), significant models were obtained comparing the plasma lipid profiles between PKU and control groups. The OPLS-DA score plots illustrate a clear discrimination between the groups (Figure 3A,B). Because the plasma lipidome is significantly different in treated PKU, we further examined which features were most discriminating.

The VIP scores—a metric summarizing the importance of each variable in driving the observed group separation—were used for variable selection. The number of discriminating features (VIP > 0.5) was 1819 in ESI(+) mode and 1816 in ESI(−) mode. A total of 272 unique differential features were found between both groups, among which 196 were elevated and 76 were lower in the plasma of patients with PKU. The top 50 of the most discriminating features/lipids according to VIP score in ESI(+) and ESI(−)are displayed in Figure 3C,D.

#### 3.3.2. Annotation and Univariate Significance of Discriminating Lipid Features

For annotation of lipidomics data, the queries of accurate mass values in compound databases provided several matches with a mass error below 5 ppm. In total, 205 out of 274 features (ESI(+) 154; ESI(−) 51) were annotated, and the identity of 69 features (ESI(+) 57 features; ESI(−) 12 features) remained unknown. Lipid features were annotated at level 2 due to a lack of standards. Appendix A present a detailed summary of lipid features, including their retention times, m/z ratio, detected adducts, annotation when available, and level of identification. Examples of obtained fragmentation patterns are provided in the Appendix A.

Alterations were predominantly noted in two key classes of lipids: glycerolipids and glycerophospholipids. Ranking the top 50 lipid species with the highest VIP scores for ESI(+), 15 positions were assigned to glycerolipid species (Figure 2). Remarkably, within this class, triglycerides (TG) were particularly discriminating (range VIP score [1.2–1.8]). The total number of carbon atoms ranged from 45 to 58. The degree of saturation varied from fully saturated up to species with 10 double bonds. Furthermore, the list included two diacylglycerol (DG) species: 34:2 [iso 3] and 40:6.

Regarding glycerophospholipids, the top 50 VIP values for ESI(+) and ESI(−) demonstrated that phosphatidylcholines (PC) and lysoPC (LPC) exhibit prominent influence within the lipidomic profile (range VIP score [0.87–1.9]). Specifically, 16 (ESI(+) and 12 (ESI(−) PCs and LPCs were in the top 50. The VIP score of multiple species exceeded 1.5, indicating a significant contribution to the predictive power of the models, including LPC: 20:3, 22:6 [iso 1], 22:6 [iso 2], P-22:6 and PC: 32:1, 36:5, 38:7 [iso 2], 42:6, O-44:4, O-44:5, O-34:2 or its isomer P-43:1. Conversely, LPC: 17:0, 18:2, 20:4, 20:5 [iso 2], 20:5 [iso 1], 22:5, LPC O-18:0, and PC: 32:2, 34:1, 36:1, 38:3, 38:7 [iso 1], 40:6, O-44:5 had a VIP score below 1.5.

Additional discriminating PL subgroups comprised lysophosphatidylethanolamines (LPE): 17:0, 17:1, 18:2, 22:0, 22:6, lysophosphatidylglycerols (LPG): 17:0, phosphatidylethanolamines (PE): 34:0, O-38:5 or P-38:4, phosphatidylinositols (PI): 34:1, 34:2, 36:1, 36:4, 39:1, 38:2, 38:3, 40:3 [iso 1], 40:3 [iso 2], O-42:6 and phosphatidylserines (PS): PS 34:0, O-38:6. Other notable lipid subgroups/species within the top 50 encompassed Ceramides (Cer): 34:1; O2 and 38:1; O2, Ceramide Phosphates (CerP): 40:1; O2, Ceramide Phosphoethanolamines (CerPE): 36:1; O2, 38:2; O2 [iso 2], Hexosylceramides (Hex2Cer): 32:0; O2, 42:2; O2, 40:1; O2, Inositol Phosphoryl Ceramides (IPC): 36:0; O2, Sphingomyelins (SM): 42:3; O2, Sulfatides (ST): 27:1; O; S Fatty Acids (FA): 16:0, 17:0, 18:3, 22:5, 22:6 and Cholesteryl Esters (CE): 18:2.

Univariate statistical testing of features with a VIP above 0.5 revealed that 33 lipid features differed statistically (adjusted *p*-value < 0.05) (Appendix A). Results are displayed in Figure 4. In total, 30 features were found to be higher, and 3 features were lower in PKU patients. More precisely, the following lipids were significantly higher in PKU patients: LPC 20:3, PI 36:4, 38:3, 40:3, PC: 32:0, 32:1, 34:1, 36:1, 36:5, 38:3, 38:6, 38:7 [iso 2], 40:6, 42:6, O-44:4, CerPE 38:2; O2 [iso 2], Hex2Cer 32:0; O2 and TG 45:5, 50:1, 50:2, 56:7, 58:7. In contrast, the following lipid species were significantly lower: CE 18:2, LPC 17:0, O-34:2/P-34:1. Furthermore, seven features without annotation were significantly higher in PKU patients compared to healthy controls. In ESI(+), the neutral masses (adducts ≥ 2) of the features were 375.2526 u, 384.1917 u, 360.1796 u, and 646.4526 u, and for ESI(−): 833.6073 u and m/z 752.5923. Boxplots of all significantly altered lipid features are shown in Appendix A.

Because of the relatively high proportion of females in the lipidomics datasets, we conducted the non-parametric Wilcoxon rank sum test within the PKU population (n = 21) to determine if sex is related to the potential biomarkers. The results show that there was no statistically significant difference in the intensity of lipidomic features between the male and female groups.

### 3.4. Metabolomics

#### 3.4.1. Multivariate Analyses of Serum Metabolomics Data

The datasets were processed according to the procedure applied to the lipidomics data. After normalization, the majority of the quality control samples clustered together, indicating the repeatability and stability of the analytical platform. It should be noted that in ESI(+), the batch effect was not fully corrected by normalization on the IS. The serum metabolome showed a separation trend in the PCA scatter plots (Appendix A), denoting a difference in metabolic profiles. The pattern identified in the score plot was further analyzed with OPLSDA to understand the relationships and differences between samples, facilitating the identification of discriminating features (Appendix A). Among them, 160 features (ESI(+) 51 features, ESI(−) 67 features) were found to be reliable for statistical interpretation based on the secondary exclusion criteria.

#### 3.4.2. Annotation and Univariate Significance of Discriminating Metabolite Features

For metabolomics, 60 out of 118 features were successfully annotated (ESI(+) 29 features and ESI(−) 31 features) (Appendix A). The identities of the remaining features are yet to be determined, comprising 22 features in ESI(+) and 35 features in ESI(−). Univariate testing revealed that 56 metabolite features (ESI(+) 13 features, ESI(−) 43 features) were significantly different (adjusted *p*-value < 0.05) compared to controls (Appendix A). Results are presented in the volcano plot (Figure 5). In total, 48 features were found to be lower (FC [0.08–0.75]), and 8 features (FC [1.15–1267]) were higher in PKU patients compared to controls. Significantly altered features were visualized using boxplots (Appendix A).

Phe and Phe catabolites were significantly higher in PKU patients, including phenyl pyruvic acid, p-hydroxyphenyl acetic acid, N-lactoyl-Phe, phenyllactic acid, glutamyl-Phe (Glu-Phe), hexanoyl-Phe (Hex-Phe), leucyl-Phe (Leu-Phe), Phe, glycyl-Phe (Gly-Phe), and phenylacetylglutamine. Furthermore, the abundance of tryptophan (Trp) derivatives indole-3-lactic acid, indole-3-carboxaldehyde, and N1-methyl-2-pyridone-5-carboxamide were also significantly higher (Figure 6).

The level of three dipeptides composed of large neutral amino acids was significantly higher, including isoleucyl-valine/leucyl-valine, tryptophan-histidine, and isoleucyl-isoleucine/leucyl-isoleucine/leucyl-leucine. Additionally, the dipeptides alanylglycine (Ala-Gly) and prolylhydroxyproline (Pro-Hyp) were significantly higher. Also, pantothenic acid, better known as vitamin B5, was higher. One annotated metabolite was significantly lower: ornithine. Metabolomics data partly confirmed alterations found with lipidomics. More specifically, there was a significantly higher level of FA 18:3; LPE 20:0 [iso 1], 20:0 [iso 2], 22:1; and LPC 20:3. In contrast, LPE 22:6 and 20:4 were significantly lower

Furthermore, there was a notable significant difference in 26 metabolite features without annotation. In more detail, one feature in ESI(+): m/z 186.1124, and three features in ESI(−): m/z 602.3458, 199.0061, and 209.0818 were lower. Likewise, a higher abundance was observed for a single feature in ESI(+): m/z 282.1195 and 21 features in ESI(−) m/z 144.0452, 143.1075, 201.0299, 339.2332, 174.0592, 151.0389, 150.0018, 329.1505, 163.0762, 147.0448, 712.2200, 91.0551, 511.2909, 385.1688, 369.1740, 383.1533, 604.3615, 187.1336, 539.2494, 165.9789, and a single feature for which a neutral mass (≥2 adducts present) was determined: 204.1360 u. With this method and the established criteria (mass error < 5 ppm), no or multiple (>3) matches were found for the identity of these features.

To determine if sex is a determinant of the potential biomarkers, we conducted the non-parametric Wilcoxon rank sum test within the PKU population (n = 35). Only the abundance of the feature P_Ile-Val/Leu-Val in the female group was statistically significantly higher than the male group (*p*-adjusted < 0.01); no other significant differences were found.

### 3.5. Correlation between Concurrent Dried Blood Spot Phe and Potential Biomarkers

In the PKU patients, concurrent dried blood spot Phe levels previously analyzed during the COBESO study (μmol/L) showed a strong significant correlation with Phe abundance of 0.88 (*p*  <  0.01). Additionally, several features demonstrated a strong significant correlation (Spearman ρ 0.50–1.00, *p* < 0.01) with concurrent Phe levels, including Glu-Phe, phenylpyruvic acid, 3.25_147.0448 m/z, phenyllactic acid, hydroxyphenylacetic acid, 4.49_151.0398 m/z, 3.25_329.1505 m/z, 7.93_91.0551 m/z, indolelactic acid, 7.40_144.0452 m/z, indole-3-carboxaldehyde, N-lactoyl Phe, phenylacetylglutamine, Ala-Gly, and 3.26_712.2200 m/z. Moderate significant correlations (0.30–0.49, *p* < 0.05) were observed with 10.71_163.0762 m/z, Trp-His, 1.94_282.1195 m/z, 4.68_174.0592 m/z, 12.05_143.1075 m/z, and 13.20_369.1740 m/z. No significant correlation was found with any of the other biomarkers. The detailed results of these correlations are presented in Spearman rank order in Appendix A.

### 3.6. Metabolic Pathway Enrichment Analysis

Annotated metabolomic features were subjected to pathway analysis. Results of the analysis are shown in Figure 7 and Table 3. Results demonstrate that the identified metabolites important for PKU were mainly representative (raw *p*-value> 0.05 and FDR *p*-value < 0.1) for the following metabolism pathways: phenylalanine metabolism, phenylalanine, tyrosine and tryptophan biosynthesis, and nicotinate and nicotinamide metabolism.

## 4. Discussion

Early-diagnosed and treated patients suffering from PKU can exhibit cognitive and/or behavioral abnormalities despite intensive treatment monitoring of Phe levels. Although blood Phe concentrations correlate with neurocognitive and behavioral outcomes, blood Phe only partly explains these outcomes [49]. This reinforces the hypothesis that other metabolites beyond Phe are involved in the pathogenesis of PKU. Identification of novel metabolites that better explain the pathophysiology underlying PKU disease could provide markers that are of diagnostic, prognostic, and/or therapeutic value. In this study, we conducted untargeted lipidomic and metabolomic analysis using high-resolution mass spectrometry platforms in a population of treated PKU patients and non-PKU controls. The PKU patients were part of the COBESO study, whose neurocognitive functioning was assessed. In this work, we were able to show that the plasma lipid and serum metabolite signature of PKU patients is different from healthy controls.

In the analysis of the metabolomics data, Phe, Phe catabolites (phenylpyruvate, phenyllactate, phenylacetate, hydroxyphenylacetic acid), Phe-conjugates (Phe-Hex, Lac-Phe, Phenylacetylglutamine) and Phe-containing dipeptides (Glu-Phe, Leu-Phe, Gly-Phe) were significantly elevated in the PKU population. The results are consistent with our previously conducted targeted analysis of this metabolomics dataset using a different bioinformatics workflow, which searched specifically for 11 previously described PKU biomarkers, including Phe, N-lactoyl-Phe, Glu-Phe, Phe-Hexose, Glu-Glu-Phe, Phe-Phe, Phe-Leu, N-acetyl-Phe, phenyllactate, phenylacetate, and phenylpyruvate [49]. Phenylacetylglutamine and Gly-Phe were not included previously, and also have not been described as altered in plasma before. However, a higher level of phenylacetylglutamine has been reported as a urinary marker for non-treated PKU patients [50,51,52].

Most Phe and Phe derivatives show a significant strong positive correlation with concurrent Phe levels. These components may seem less interesting as new biomarkers because the increases in the Phe-containing metabolites are inherent to elevated Phe levels in the blood and probably the intake of balanced amino acid supplements. However, Wegberg et al. [49] demonstrated that N-lactoyl-Phe outperformed Phe as a predictor of working memory and mental health outcomes. These findings suggest that N-lactoyl-Phe may be a better biomarker to reflect neurocognitive outcomes than Phe. Additionally, phenylacetylglutamine is produced from phenylacetic acid and Phe, which may competitively inhibit L-amino acid decarboxylase, thereby decreasing neurotransmitter levels [53]. Regarding Phe-containing dipeptides, limited information exists about their general physiological significance or their clinical and pathophysiological relevance for PKU.

The level of pantothenic acid, also referred to as vitamin B5, was reported to be higher in the PKU population. This micronutrient is a precursor of coenzyme-A (CoA) and the acyl carrier protein. It is involved in cellular energy production and the synthesis of carbohydrates, proteins, and fats. Further, it is involved in the biosynthesis of essential lipids, steroids, hormones, neurotransmitters, and porphyrin, and the formation of red blood cells, sex, and stress-related hormones. Stroup et al. [54] reported that without supplementation, PKU patients are at risk of inadequate intake of pantothenic acid. Recently, Bokayeva et al. [55] pointed out the lack of publications describing vitamin status, including vitamin B5, in PKU patients and the effect on clinical outcomes. We speculate that the higher level of vitamin B5 in our study is most likely a result of its presence in supplements.

Interestingly, multiple Trp derivatives were found to be significantly higher in the serum of PKU patients, including indole-3-lactic acid (I3LA), indole-3-carboxaldehyde (I3A), and N1-methyl-2-pyridone-5-carboxamide (2PY). Moreover, N1-methyl-4-pyridone-3-carboxamide (4PY) and indole-3-propionic acid (IPA) levels were higher in PKU patients and almost reached statistical significance (adjusted *p*-value = 0.05). Trp is an essential amino acid, meaning that it is obtained from the diet or protein catabolism but not synthesized endogenously. Trp itself is a precursor for various biologically active metabolites, including the neurotransmitter serotonin. The gut mucosa’s enterochromaffin cells (ECs) are primarily responsible for the body’s serotonin synthesis. Serotonin is released by activated endothelial cells (ECs) into the interstitium of the lamina propria, where it binds intrinsic sensory neurons’ nerve endings [53]. Being part of the microbiota–gut–brain axis, Trp and its metabolites could be of interest for clinical outcomes in PKU.

Trp is metabolized along four pathways, including serotonin (serotonin and melatonin), tryptamine (tryptamine), indole (indolepyruvic acid), and kynurenine (niacin) pathways. The majority (~90%) of Trp is metabolized through the kynurenine pathway. The metabolites that were found to be elevated in our PKU patients are downstream catabolites of the kynurenine (2PY, 4PY) and indole (I3LA, I3A, IPA) pathways (see also Figure 6). Higher levels of 2PY and 4PY have been described before in urine samples of early-treated PKU patients [56]. The reason for this relation and the clinical relevance is uncertain. Both compounds are considered to be uremic toxins based on the fact that their concentrations increase with decreased kidney function and based on their ability to inhibit poly(ADP-ribose) polymerase-1 (PARP1). This enzyme is involved in cellular response to DNA damage [57,58,59,60]. Both compounds are, however, end products of Trp catabolism (kynurenine pathway) via the degradation of nicotinamide-adenine dinucleotide (NAD). In theory, higher levels might merely reflect an increased Trp and/or niacin status or a rerouting of Trp metabolism in the kynurenine pathway. The niacin status of PKU patients is not well described. The implications of this finding need further research.

Several studies suggest changes in the gut bacterial microbes of PKU patients who have undergone treatment compared to individuals without PKU. In more detail, PKU patients exhibited a higher number of Trp-consuming bacteria in the colon [61,62]. Alterations in bacteria consuming Trp might affect serotonin synthesis in the brain since it modulates the Trp availability in the circulation. Only I3A has been detected before in the urine of patients with untreated PKU. While more research is needed, these findings support those described by Parolisi et al. [63,64], highlighting the relevance of exploring the potential effects of Trp availability through gut microbiota on cognitive and behavioral functions in PKU.

We observed that PKU patients exhibit a different lipid signature compared to healthy individuals. Differences were predominantly observed between lipids belonging to the glycerophospholipid and glycerolipid classes. This observation confirms that closer evaluation of the lipidome in PKU patients might be interesting, as previously emphasized by Guerra et al. [65,66,67,68]. We identified a significant increase in various glycerophospholipids, including (L)PE, (L)PC, and PI. Most species incorporated at least one fatty acid with an unsaturated aliphatic chain, as indicated by the total number of carbons and double bonds. None of the features were correlated with concurrent Phe, which makes them potential biomarkers of interest for PKU, reflecting either pathophysiological processes or consequences of the dietary treatment.

Although we were unable to elucidate the full molecular structure of the fatty acyl chain(s), previous studies and data provided herein provide a level of insight. To date, alterations in plasma-free FA status, particularly in n-3 FA were observed in PKU patients treated with a Phe-restricted diet [69,70,71,72,73]. The n-3 long-chain PUFA species DHA (22:6 n-3) and eicosapentaenoic acid (EPA, 20:5n-3) were consistently demonstrated to be reduced [70,72,74,75]. These findings were attributed to dietary restrictions. PKU patients are advised to consume no or only a limited amount of animal products because of the high protein content in these products. As a result, foods rich in saturated fatty acids and long-chain PUFAs are excluded from their diet [74,76]. Some protein substitutes are enriched with DHA, but not all of them. Long-chain PUFAs, particularly arachidonic acid (AA, 20:4n-6) and DHA, are crucial for normal neurological development [77,78,79,80]. Based on these findings, it is recommended to supplement EPA and DHA if these are not already added to the protein substitute [17]. Unfortunately, dietary data were not available for this cohort. Studies show that suppletion increases the free and PL plasma levels of these FAs in PKU [81,82,83,84,85]. Although there is a debate whether the difference is clinically relevant it is known to improve fine motor skills and short-term visual response in children with PKU [69,70,74,81,82,83,84,85,86,87].

Furthermore, the FA composition of PL-enriched plasma extracts has been reported in several studies. For instance, Drzymała-Czyż et al. [88] reported a reduction in FA: 20:3n-9, 18:2n-6 (LA, linoleic acid), and 22:6n-3 (DHA). In 2021, Guerra et al. [89] investigated the plasma PL and sphingolipid profile in children treated for PKU. In short, they found a significant increase in FA 18:1n-9 (oleic acid), 14:0 (myristic acid), 18:0 (stearic acid), EPA, and DHA, while no significant changes in statistical difference were observed for FA 16:0, LA, and AA. Consistent with these findings, our detailed analysis of PL species confirmed a significant increase in PC species (32:0, 32:1, 34:1, 36:5, 38:7, 40:6, and 42:6) and PI species (36:4 and 38:3). Altogether, changes in FA status due to the PKU diet may explain the observed alterations in the PL profile of PKU patients. Additional measurements to fully elucidate the structure of the FA chains are essential to give further meaning to these findings.

The total triglyceride (TTG) concentration was significantly elevated in PKU patients. This only captures a fraction of the total lipid content and provides limited detailed information about the level of individual species. To the best of our knowledge, this is the first study that described detailed information about TG alterations. In addition, we observed a significant increase in various TG (45:5, 50:1, 50:2, 56:7, and 58:7) containing saturated and unsaturated fatty acyl chains. To date, some perturbations in lipoprotein components of PKU individuals have been described in plasma and serum. However, the results are contradictory. To demonstrate, four studies report a reduction in TC, LDL-C, and/or HDL-C [90,91,92,93]. However, Azabdaftari et al. [94] reported an increase in TC content in serum. Similarly, VLDL-C has been found to be higher [90,95] but also reduced [92,94]. Studies in both plasma and serum show that total TG levels increase in children and older PKU patients [90,92,95,96,97]. This phenomenon was associated with increased carbohydrate intake.

The implications of higher TTG levels for clinical outcomes are not fully understood. However, it is known to correlate with the accumulation of visceral fat, which, in turn, is linked to a higher risk of developing type 2 diabetes and cardiovascular diseases [98,99]. This might also be relevant for PKU patients, as there are reports of obesity, dyslipidemia, hypertension, and oxidative stress in PKU patients [91,100,101,102,103]. Furthermore, in the present study, we found an increase in CerPE 38:2. CerPEs are produced in small amounts and located in the plasma membrane. To date, their function remains largely unknown [104]. However, higher levels of CerPE 38:2 have been suggested to increase the risk of type 2 diabetes [105]. Additionally, we report lower levels of LPC 17:0, which is also known to be inversely correlated with type 2 diabetes [106,107]. Higher TTG and CerPE 38:2 abundance and lower levels of LPC 17:0 might suggest that this cohort of PKU patients is at increased risk of CVD and type 2 diabetes. However, the question remains whether this risk is comparable to that of non-PKU individuals. Unfortunately, no anthropometric data were available for our study cohort. The link between CVD or diabetes and the plasma lipid composition may be of interest to future research.

Strengths of this study include a comprehensive metabolomic and lipidomic analysis of plasma/serum obtained in a well-defined population of PKU patients. Patients who participated in the COBESO study are well described with respect to their neuropsychological outcomes [24,25,26,27]. We are, therefore, able to correlate the biomarkers to clinically relevant outcomes. We chose to comprehensively describe the analytical methods and important biomarkers for PKU in this study, whereas we aim to describe the relation with neuropsychological outcomes in a separate paper, allowing both topics to receive the attention and depth they deserve. Another advantage of our combined untargeted-omics approach is that we found several new PKU biomarkers; in particular, the lipidome is rather unexplored. Further studies are necessary to explore the relevance of our findings.

A limitation of our study is that the control and patient samples were obtained from different studies, potentially introducing variability. We would like to emphasize that this study is exploratory and acknowledge that variables such as long-term sample storage may have affected the metabolome or lipid signature. It is well known that long-term storage significantly impacts the metabolome and the lipidome [108,109], including amino acids such as Phe [110,111], although the impact of storage of EDTA-plasma on the metabolome seems limited when samples are stored up to 7 years at −80 degrees Celcius [112]. Unfortunately, no long-term data on the stability of the PKU biomarkers found in our study are available in our laboratories. The impact of the difference in storage time between PKU patients and the non-PKU controls on the biomarkers is, therefore, uncertain. On the other hand, some of the biomarkers have been found by other groups as well, demonstrating the validity of these biomarkers. Another important aspect is that our results were obtained in a relatively small population. In combination with the other potential confounding variables, we realize that any conclusions based on this work are to be considered preliminary and should be further explored.

In the future, we intend to conduct correlation analyses between the biomarkers identified in this study and the previously obtained neurocognitive outcomes from the COBESO study. If these analyses reveal any interesting results, we will proceed to confirm the identity of the detected metabolomics and lipidomics features of interest. We intend to confirm the identity, preferably using standards, in addition to performing a DDA (data-dependent acquisition) experiment, allowing us to generate clean spectra and making feature annotation more reliable. Naturally, this study must be repeated with more recent data to verify the findings, preferably in a larger prospective study applying a quantitative analytical method, avoiding long-term storage effects. Additionally, future studies should incorporate anthropometrical and, most importantly, nutritional metrics to contextualize the metabolomic and lipidomic findings to diet.

To summarize, in this study, we have shown that the lipidomic and metabolic profiles of treated adult PKU patients differ from non-PKU individuals. Regarding lipidomics, total TG content was higher and alterations were found for lipid features belonging to the classes of glycerolipids, glycerophospholipids, and sphingolipids. Metabolomic results confirmed alterations in the lipidome. Furthermore, metabolomic results demonstrated that patients had higher levels of Phe and Phe catabolites, including N-lactoyl-Phe, Trp derivatives, pantothenic acid, and various dipeptides. Future studies should focus on the identification of unknown features and correlate significantly different lipid and metabolite features with outcome parameters. Altogether, this research provides insight into the metabolomic and lipid status of treated adult PKU patients. However, the clinical and/or therapeutic relevance and validity of the biomarkers need to be determined.

## Figures and Tables

**Figure 1 metabolites-14-00479-f001:**
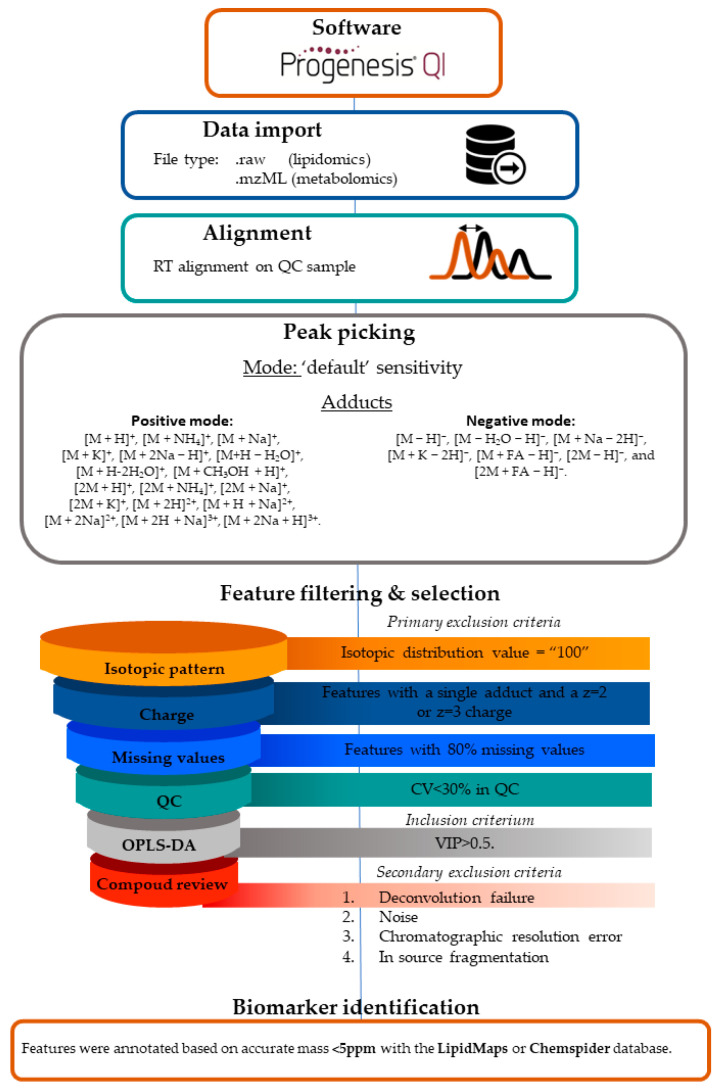
Flowchart of the data processing workflow. This figure illustrates the workflow applied to the datasets in this study to identify potential biomarkers using the Progenesis QI software Version 2.4. The process is divided into several steps, including data import, retention time alignment, peak picking, feature filtering and selection, and biomarker identification. Details of the workflow are extensively described in Appendix A.

**Figure 2 metabolites-14-00479-f002:**
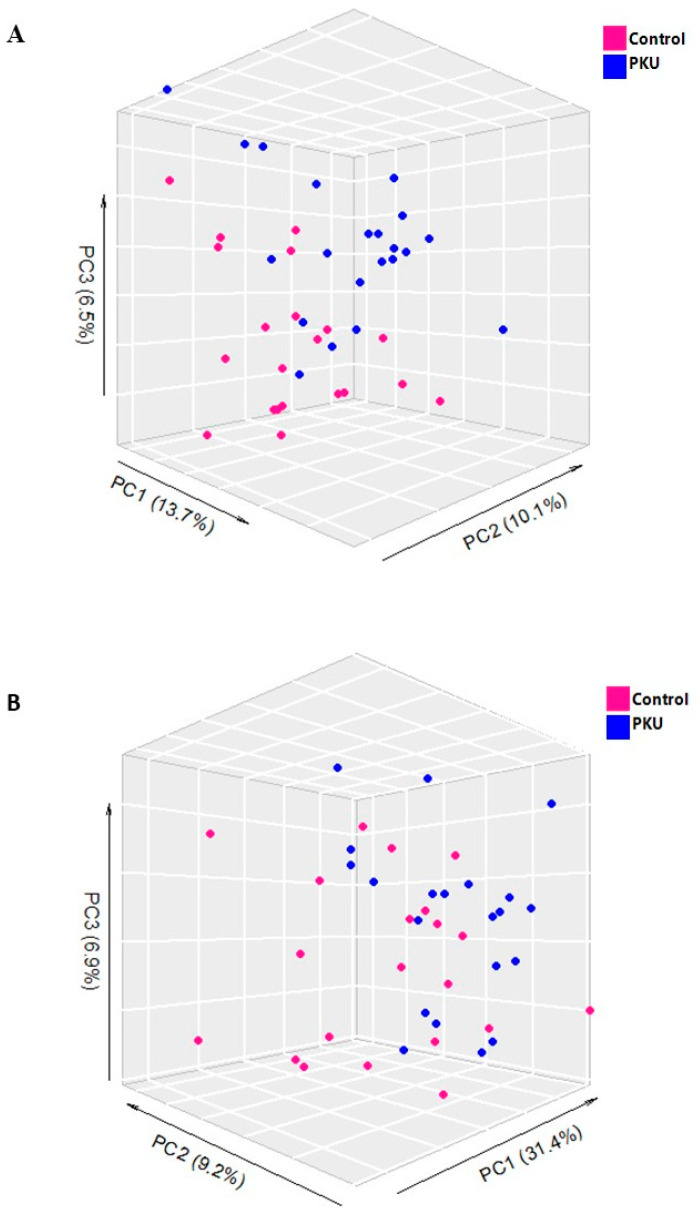
Three-dimensional PCA score plots of lipidomic profiles from ESI(+) and ESI(−) modes for control and PKU patients. Principal Component Analysis (PCA) score plots derived from lipidomic data obtained in both ESI(+) (**A**) and ESI(−) (**B**) ionization modes. The plots illustrate the spatial distribution of controls (n = 21) and PKU patients (n = 21), represented by blue and red circles, respectively. The axes of the plots correspond to the first three principal components (PC1 on the *x*-axis, PC2 on the *y*-axis, and PC3 on the *z*-axis), which capture most of the variances within the dataset. Panel (**A**), corresponding to the ESI(+) mode, reveals an R^2^X value of 0.50 and a cumulative Q^2^ of 0.19, indicating a moderate explanation of variance and predictability from the model. Meanwhile, Panel (**B**), illustrating ESI(−) mode, demonstrates a higher model efficacy with an R^2^X of 0.66 and a cumulative Q^2^ of 0.40.

**Figure 3 metabolites-14-00479-f003:**
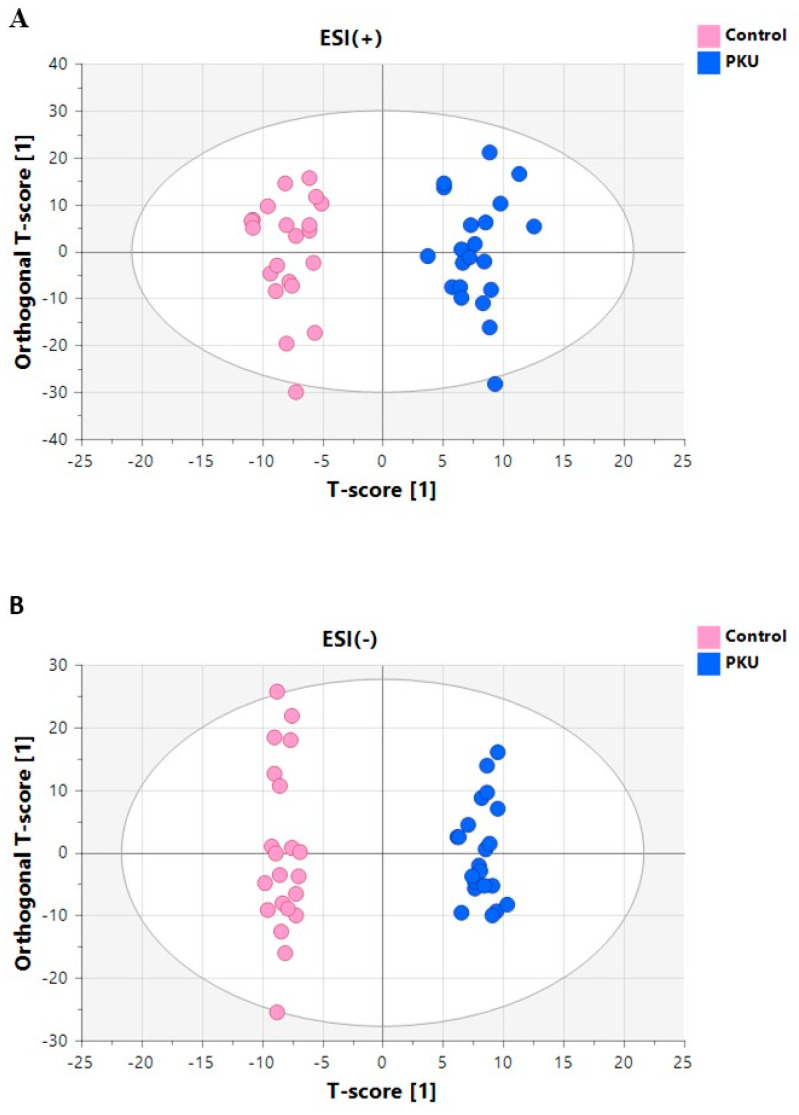
OPLS-DA results of lipidomic profiles in PKU patients compared to controls. The figure displays OPLSDA score plots for pairwise comparisons between PKU and control groups, utilizing lipidomics data. Panels (**A**,**B**) illustrate the results for ESI(+) and ESI(−), respectively. Each point represents the lipidomic profile of an individual, with PKU patients and controls plotted to show the clear distinction and grouping based on their lipid abundances. For ESI+, the model diagnostics are as follows: R^2^X = 0.249, R^2^Y(cum) = 0.943, and Q^2^ = 0.42, indicating a strong model with significant predictive capability and reliability in distinguishing between PKU patients and controls. The colored circles represent 95% confidence regions, demonstrating tight clustering and clear separation between the groups. In the negative ionization mode (**B**), the model shows a higher complexity and fit with R^2^X = 0.52, R^2^Y(cum) = 0.986, and Q^2^ = 0.673. Both models were statistically significant, with *p*-values below 0.05, underscoring the robustness of the lipidomic differences identified. Additionally, panels (**C**,**D**) display the top 50 features with the highest VIP scores for both models in ESI(+) and ESI(−), respectively.

**Figure 4 metabolites-14-00479-f004:**
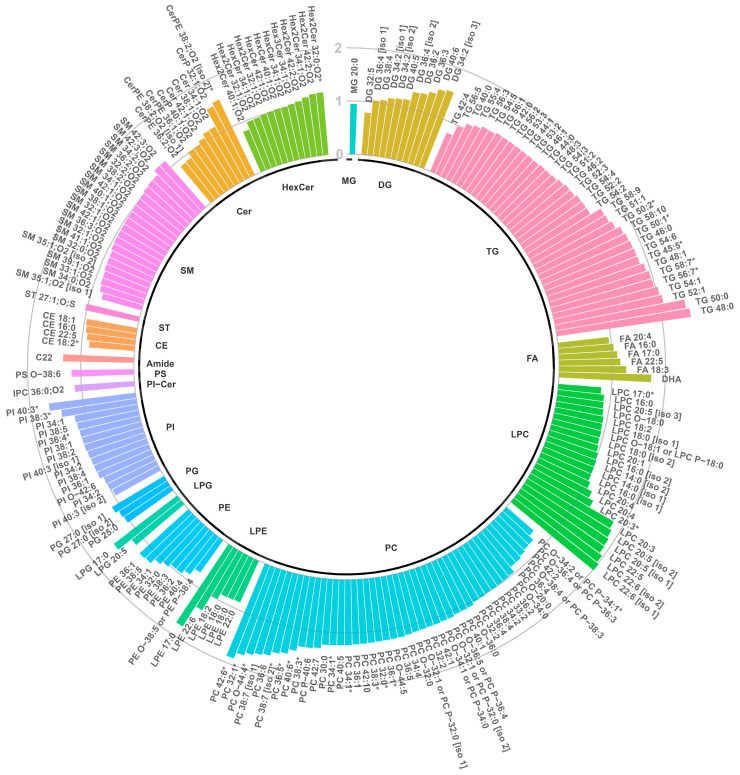
Circular bar chart expressing the fold change of annotated lipid features. The fold change is calculated as the average ratio of lipid abundance in PKU patients to that in control subjects. Bars extending outward from the circle center represent the fold change ranging from 0 to 2, indicating higher (FC > 1) or lower (FC < 1) lipid abundances in PKU patients compared to controls. Significantly higher or lower (adjusted *p*-value < 0.05) lipids are marked with an asterisk (*). The ‘O-’ prefix is used for plasmanyl species to indicate the presence of an alkyl ether substituent, whereas the ‘P-’ prefix is used for plasmenyl species to indicate the alk-1-enyl ether substituent.

**Figure 5 metabolites-14-00479-f005:**
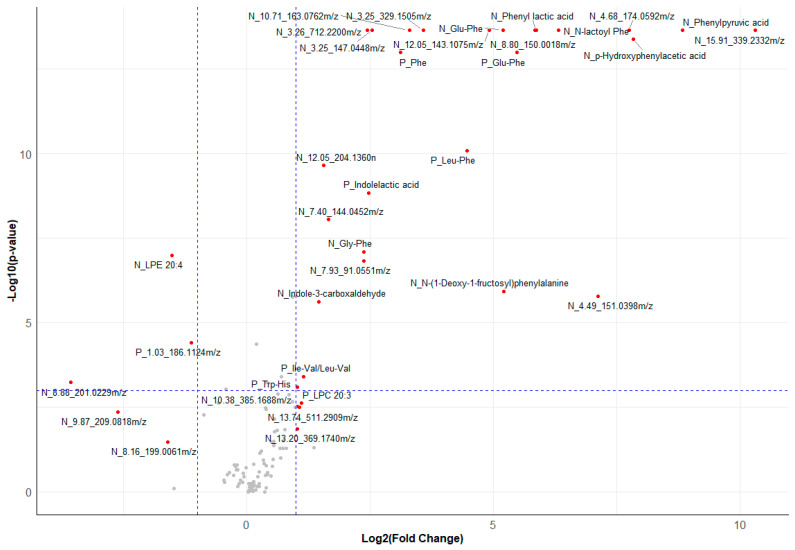
Volcano plot displaying metabolomic data. Labeled points indicate metabolite or metabolite feature is significant. Metabolites starting with ‘N_’ were detected in negative ion mode electrospray ionization (ESI−), while those with ‘P_’ were identified in positive ion mode electrospray ionization (ESI+). The x-axis represents the log2 of the fold change, and the y-axis indicates the negative log10 of the adjusted *p*-values, with the dashed lines marking thresholds for statistical significance.

**Figure 6 metabolites-14-00479-f006:**
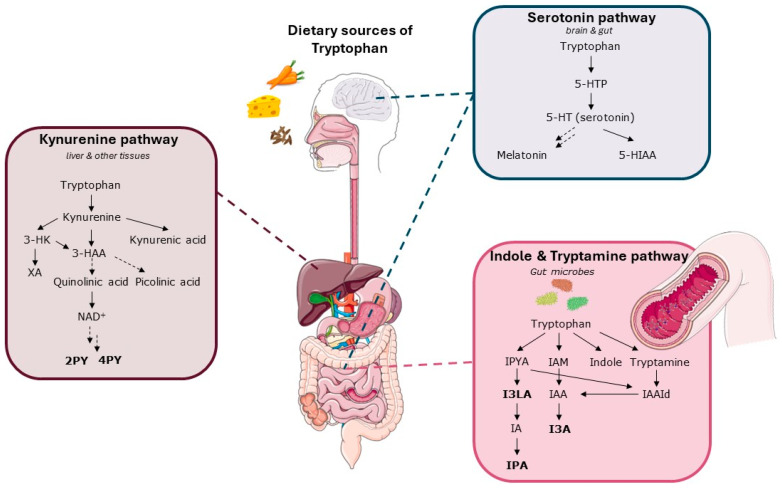
Overview of detected metabolites and their location in the Trp metabolism. Simplified representation of the metabolic pathways of Trp in the human body. Trp is an essential amino acid that is catabolized by four key metabolic routes, including the serotonin, kynurenine, indole, and tryptamine pathway. The pathways have been thoroughly reviewed previously [44,45]. In short, after food consumption, most of the Trp is absorbed in the small intestine and enters the bloodstream bound to albumin or as free Trp. The remaining 4–6% of Trp reaches the large intestine and is the primary precursor of indole, tryptamine, and other molecules derived from the microbiome [46]. The indole and tryptamine pathway involves microbes, for example, bacteria belonging to Bacteroides, Clostridium, and Bifidobacterium. The kynurenine pathway is the primary route for Trp degradation in the liver. Trp is converted to kynurenine by the enzyme tryptophan 2,3-dioxygenase (TDO) in the liver or indoleamine 2,3-dioxygenase (IDO) in other tissues [47]. Kynurenine can then be further metabolized into several bioactive compounds, including kynurenic acid and quinolinic acid. This route is also the starting point for the synthesis of nicotinamide adenine dinucleotide (NAD) and other downstream metabolites, including 2PY and 4PY. In the serotonin pathway, the enzyme tryptophan hydroxylase (TPH) is predominantly expressed in the brain (TPH2) and enterochromaffin cells in the gut (TPH1) [48]. They convert Trp to 5-hydroxytryptophan (5-HTP), and thereafter, 5-HTP is converted to serotonin (5-hydroxytryptamine, 5-HT) that enters the circulation. Significantly elevated metabolites (2PY, I3LA, I3A) or almost reached significance (IPA, 4PY) are marked bold. 3-HAA = 3-hydroxyanthranilic acid, 5-HIAA = 5-hydroxyindoleacetic acid, 3-HK = 3-hydroxykynurenine, 5-HT = 5-hydroxytryptamine, 5-HTP = 5-hydroxytryptophan, IA = 3-indoleacrylic acid, IAA = indole-3-acetic acid, IAAld = indole-3-acetaldehyde, I3A = indole-3-carboxyaldehyde, IAM = indole-3-acetamide, I3LA = indole-3-lactic acid, IPA = indole-3-propionic acid, IPYA = indole-3-pyruvic acid, NAD = nicotinamide adenine dinucleotide, XA = xanthurenic acid.

**Figure 7 metabolites-14-00479-f007:**
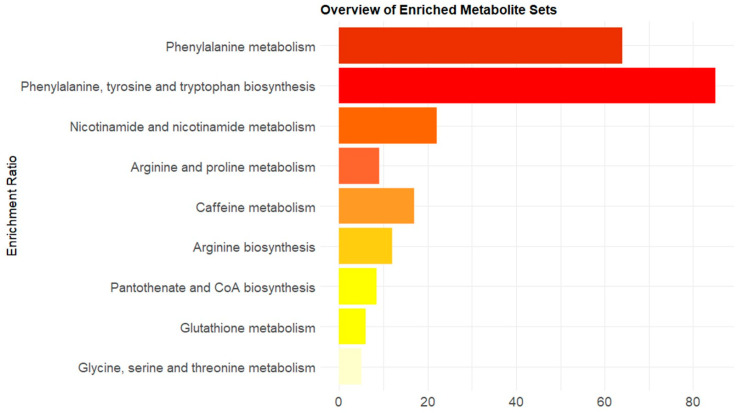
Results of the pathway enrichment analysis. The enrichment ratio reflects the number of hits divided by the expected number of hits. The pathways are ordered from low to high raw *p*-value.

**Table 1 metabolites-14-00479-t001:** Characteristics of the study population ^1^.

	Metabolomics		Lipidomics	
	PKU	Control	PKU	Control
n	35	20	22	22
Age (years)	29 ± 7	34 ± 5	33 ± 6	34 ± 5
Sex				
male	18 (51%)	12 (60%)	8 (36%)	8 (36%)
female	17 (49%)	8 (40%)	14 (64%)	14 (64%)
Lifetime Phe (µmol/L)	439 [223–1001], n = 33	-	446 [255–734], n = 20	-
Concurrent Phe (µmol/L)	616 [176–1250], n = 35		645 [176–1250], n = 22	
BH4 treatment	8/35	-	2/22	-
Plasma total cholesterol (mmol/L)			4.39 ± 0.81	4.65 ± 0.66
Plasma total triglyceride (mmol/L)			1.52 ± 0.55	1.32 ± 0.74
Matrix for the current study	Serum	Serum	EDTA-plasma	EDTA-plasma

^1^ Results are expressed as mean ± standard deviation (SD) or as median [range].

**Table 2 metabolites-14-00479-t002:** The number of features retained after each filtering step.

	Lipidomics		Metabolomics			
	ESI(+)	ESI(−)	ESI(+)		ESI(−)	
			Batch A	Batch B	Batch A	Batch B
Number of features detected	15,220	5223	3627		2258	
Primary exclusion criteria						
An isotopic abundance of 100	3570	527	1291		571	
Predominant adduct with z = 2 or 3	3285	0	157		0	
Features with 80% missing values	1320	12	314	430	422	314
CV > 30%	2607	568	923	1145	507	562
Number of features passing primary criteria	4438	4416	725		503	
OPLS-DA						
VIP > 0.5	3226	3243	628		281	
Secondary exclusion criteria after OPLS-DA						
The final number of unique features	210	64	51		67	

**Table 3 metabolites-14-00479-t003:** Detailed results from the pathway enrichment analysis.

	Total ^a^	Expected	Hits ^b^	Raw *p* ^c^	Holm *p* ^d^	FDR ^e^
Phenylalanine metabolism	8	0.047	3	7.64 × 10^−6^	0.000612	0.000612
Phenylalanine, tyrosine and tryptophan biosynthesis	4	0.023	2	0.000181	0.0143	0.00726
Nicotinate and nicotinamide metabolism	15	0.088	2	0.00307	0.239	0.0819
Arginine and proline metabolism	36	0.211	2	0.0173	1	0.346
Caffeine metabolism	10	0.059	1	0.0571	1	0.914
Arginine biosynthesis	14	0.082	1	0.0792	1	1
Pantothenate and CoA biosynthesis	20	0.117	1	0.111	1	1
Glutathione metabolism	28	0.164	1	0.153	1	1
Glycine, serine and threonine metabolism	33	0.193	1	0.178	1	1

^a^ Total: the number of metabolites in the pathway ^b^ Hits is the matched number of metabolites from the data ^c^ Raw *p*: is the original *p*-value calculated from the enrichment analysis ^d^ Holm *p*: is the *p*-value adjusted by the Holm–Bonferroni method ^e^ FDR: *p* is the *p*-value adjusted using false discovery rate.

## Data Availability

The data presented in this study are available on request from the corresponding author. The data are not publicly available due to privacy and ethical restrictions.

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
