# Peer review of "Impact of Phenylketonuria on the Serum Metabolome and Plasma Lipidome: A Study in Early-Treated Patients"

_metabolites, 2024, doi:10.3390/metabo14090479_

Round 1

Reviewer 1 Report

Comments and Suggestions for Authors

The work enclosed in the manuscript was carried out with competencies and coupling lipidomics with metabolomics signatures is an interesting approach. However, some clarifications on the work done are necessary. Below my minor comments of the work.

-it is relevant that the number of patients considered is small and can be a weakness of the paper. Anyway, the authors should better describe the utility of the findings for the future research and how this data can help scientists in the field.

-novelty of this study related to the previous performed in finding biomarkers for PKU should be better highlight

-I noted a significant difference in the number of male and female. There is a gender bias? Furthermore, the signatures are different in male and female? possible biomarkers could be consider gender-dependent? a discussion in this sense should be useful.

Comments on the Quality of English Language

minor

Author Response

Comments 1: it is relevant that the number of patients considered is small and can be a weakness of the paper. Anyway, the authors should better describe the utility of the findings for the future research and how this data can help scientists in the field.

Response 1: Thank you for your comments on our manuscript. We acknowledge that the limited number of patients is a limitation of our study. However, it is important to emphasize that while inherited metabolic disorders are common collectively, they are individually rare, as is the case with PKU. This COBESO population was particularly interesting, because their neuropsychological outcome is well described, moreover, findings of some biomarkers correlate with those found in other studies, which may be considered a reflection of the validity of our findings. To accommodate your request, we have conducted additional analyses, including correlation analyses of the biomarkers with concurrent phenylalanine levels and their mapping in pathway analyses, see pg 19/20, lines 616-645 of the results section.

Comment 2: novelty of this study related to the previous performed in finding biomarkers for PKU should be better highlight.

Response 2: Thank you for your comment. At several points in the discussion, the novelty of some biomarkers was addressed, we nevertheless agree that some text should be added on the main advantage of our study, i.e. the possibility to correlate our findings to clinically relevant outcomes. We added a few lines addressing this in the discussion section, see page 25, lines 849-882.

Comment 3: I noted a significant difference in the number of male and female. There is a gender bias? Furthermore, the signatures are different in male and female? possible biomarkers could be consider gender-dependent? a discussion in this sense should be useful.

Response 3: Thank you. We agree that sex may be correlated to the biomarker abundances, although it is less a confounder in our study as the male/female ratio was not significantly different between the control and PKU cohorts. Nevertheless, to determine if sex is related to the potential biomarkers, we conducted the non-parametric Wilcoxon rank sum test within the PKU population. For metabolomics, the population included 17 females and 18 males. For lipidomics, the population comprised 7 males and 14 female subjects. The results show that there was no statistically significant difference in the intensity of most metabolomic and lipidomic features between the male and female groups. However, the abundance of the feature P_Ile-Val/Leu-Val in the female group was statistically significantly higher than the male group (p-adjusted <0.01). This finding does not invalidate the results of our study, as this is not considered to be the most promising biomarker of PKU. We added the results of this analysis to the revised manuscript, see pg 16, lines 507-512 and pg 18, lines 579-583 of the results section.

Reviewer 2 Report

Comments and Suggestions for Authors

Dear Authors,

Thank you for submitting your manuscript "Impact of phenylketonuria on the plasma metabolome and lipidome. A study in early treated patients." I have carefully reviewed your work and find it to be a valuable contribution to our understanding of the metabolic profiles in treated PKU patients. The use of untargeted lipidomics and metabolomics approaches provides novel insights into the complex metabolic alterations associated with this condition. However, I have several suggestions for improvement that I believe will strengthen the manuscript:

Major Revisions:

1. Sample storage and potential impact: While you acknowledge the potential effects of long-term sample storage, this issue requires more detailed discussion. Please provide information on how long the samples were stored before analysis and discuss any measures taken to assess or mitigate storage-related degradation. Consider including a brief analysis of quality control samples to demonstrate the stability of metabolites over time.

2. Control group selection: The use of pre-transplant kidney donors as controls, while pragmatic, introduces potential confounding factors. Please provide a more thorough justification for this choice and discuss any potential limitations or biases it may introduce. Consider performing a sensitivity analysis with a subset of controls more closely matched to the PKU patients, if possible.

3. Clinical correlations: The study would be significantly strengthened by correlating the observed metabolic changes with clinical parameters in the PKU patients. If available, please include data on factors such as dietary compliance, neurocognitive outcomes, or other relevant clinical measures. This would help contextualize the metabolomic findings and enhance their potential clinical relevance.

4. Pathway analysis: While you discuss individual metabolites and lipids, a more comprehensive pathway analysis would provide greater insight into the biological processes affected in PKU. Consider using tools like MetaboAnalyst to identify significantly altered pathways and include a figure illustrating these changes.

5. Validation of key findings: For the most significant metabolite and lipid alterations, consider validating a subset using targeted, quantitative methods. This would increase confidence in the untargeted results and provide more precise quantification of the changes observed.

Minor Revisions:

1. Statistical analysis: Provide more detail on the multiple testing correction methods used, particularly for the univariate analyses. Clarify whether the p-values reported are raw or adjusted for multiple comparisons.

2. Abbreviations: Ensure all abbreviations are defined at first use. For example, "TTG" is used before being defined as "total triglyceride."

3. Discussion: The discussion of tryptophan metabolites is interesting but could benefit from a brief explanation of why these changes might be relevant in PKU. Consider discussing potential implications for neurotransmitter metabolism or gut-brain axis function.

Overall, this is a well-conducted study that provides valuable insights into the metabolic landscape of treated PKU patients. Addressing these points will further enhance the scientific rigor and impact of your work. I look forward to reviewing a revised version of this manuscript.

Comments on the Quality of English Language

Minor editing of English language required

Author Response

Reviewer 2                                                                                                                                                    

Major Revisions:

Comment 1: Sample storage and potential impact: While you acknowledge the potential effects of long-term sample storage, this issue requires more detailed discussion. Please provide information on how long the samples were stored before analysis and discuss any measures taken to assess or mitigate storage-related degradation. Consider including a brief analysis of quality control samples to demonstrate the stability of metabolites over time.

Response 1: We are aware that extended storage and freeze-thaw cycles of samples may affect the concentration of metabolomic and lipidomics features. We are unfortunately unable to predict storage effects on the PKU biomarkers found in our study. We addressed this issue in the discussion, see pg 25, lines 862-874.

Comment 2: Control group selection: The use of pre-transplant kidney donors as controls, while pragmatic, introduces potential confounding factors. Please provide a more thorough justification for this choice and discuss any potential limitations or biases it may introduce. Consider performing a sensitivity analysis with a subset of controls more closely matched to the PKU patients, if possible.

Response 2: Thank you. We confirm that it was a pragmatic choice.

Our control population is not a perfectly random sample of the general population; however, which random sample truly is? Invariably, biases arise due to the methodologies employed in issuing invitations and the varied responses to these invitations. Such biases are unavoidable.

In this particular instance, the individuals involved typically have some association with patients suffering from renal failure—be it through familial ties, friendship, or acquaintance, or through altruistic motives. Furthermore, they are generally pre-screened by either a general practitioner or a peripheral nephrologist before being referred for additional evaluation. This screening process includes ensuring that they possess normal kidney function and often verifies the presence of two kidneys, which can be straightforwardly confirmed via ultrasound. We therefore expect that this population reflects a random sample of controls with the sole selection of having a normal kidney function. A short sentence was added in the materials and methods section to address this point, see pg 3, lines 131-133.

Comment 3: Clinical correlations: The study would be significantly strengthened by correlating the observed metabolic changes with clinical parameters in PKU patients. If available, please include data on factors such as dietary compliance, neurocognitive outcomes, or other relevant clinical measures. This would help contextualize the metabolomic findings and enhance their potential clinical relevance.

Response 3: Thank you for your comment. In this paper we chose to comprehensively describe the analytical methods and important biomarkers for PKU in a treated population. In a second paper, we aim to describe the relation between these biomarkers and clinically relevant outcomes, most importantly neuropsychological outcomes, allowing both topics to receive the attention and depth they deserve. We are unfortunately unable to include data on dietary compliance and anthropometrics, as these data were not collected in this study. We added a sentence on these aspects in the discussion, see and pg 25, lines 851-857, pg 26, lines 880-882.

Based on your suggestion, we performed additional analyses on the biochemical data, including the correlation with the biomarkers and concurrent phenylalanine levels, the results are integrated in the results section of the revised manuscript, see pg 19/20, lines 616-645 of the results section.

Comment 4. Pathway analysis: While you discuss individual metabolites and lipids, a more comprehensive pathway analysis would provide greater insight into the biological processes affected in PKU. Consider using tools like MetaboAnalyst to identify significantly altered pathways and include a figure illustrating these changes.

Response 4: Thank you for this valuable suggestion, we performed a pathway analysis and incorporated the results section of the revised manuscript, see pg 19/20, lines 616-645 of the results section.

  1. Validation of key findings: For the most significant metabolite and lipid alterations, consider validating a subset using targeted, quantitative methods. This would increase confidence in the untargeted results and provide more precise quantification of the changes observed.

Response 5: Thank you for your comment. We agree that targeted analysis would increase confidence in the untargeted results. For some metabolites described in this paper such as Phe-hexose, the identity was elucidated and stability studies were performed (Outersterp, 2021). In a follow-up study, we want to confirm the identity and stability of the proposed markers using a more suitable machine. Unfortunately, for this publication, we are unable to perform additional measurements regarding the full structural elucidation. However, as indicated in the manuscript some metabolites are included in an in-house database and those are confirmed with a standard, including confirmation of retention times. In this publication, this is indicated by the level of annotation.

van Outersterp, R. E., Moons, S. J., Engelke, U. F. H., Bentlage, H., Peters, T. M. A., van Rooij, A., Huigen, M. C. D. G., de Boer, S., van der Heeft, E., Kluijtmans, L. A. J., van Karnebeek, C. D. M., Wevers, R. A., Berden, G., Oomens, J., Boltje, T. J., Coene, K. L. M., & Martens, J. (2021). Amadori rearrangement products as potential biomarkers for inborn errors of amino-acid metabolism. Communications biology, 4(1), 367. https://doi.org/10.1038/s42003-021-01909-5

Minor Revisions:

  1. Statistical analysis: Provide more detail on the multiple testing correction methods used, particularly for the univariate analyses. Clarify whether the p-values reported are raw or adjusted for multiple comparisons.

“ Thank you for your comment. We clarified which p-values are reported in the manuscript.”

  1. Abbreviations: Ensure all abbreviations are defined at first use. For example, "TTG" is used before being defined as "total triglyceride."

“Thank you for your valuable feedback. We made the necessary revisions to ensure that all abbreviations are clearly defined upon their first use.”

  1. Discussion: The discussion of tryptophan metabolites is interesting but could benefit from a brief explanation of why these changes might be relevant in PKU. Consider discussing potential implications for neurotransmitter metabolism or gut-brain axis function.

“Thank you for this suggestion, we added a few sentences on this topic in the discussion section of the revised manuscript. see pg 23, lines 730-739.”

Round 2

Reviewer 2 Report

Comments and Suggestions for Authors

Dear Authors,

Thank you for your thorough responses to the reviewer's comments. After reviewing your replies, I have some suggestions for minor revisions that could further enhance the clarity and completeness of your manuscript:

1. Sample storage: While you address long-term storage effects in the discussion, consider briefly stating the specific storage duration and conditions in the methods section.

2. Control group: Please add a paragraph in the methods describing the characteristics of the control group (e.g., age range, gender ratio) to facilitate comparison with the PKU patient cohort.

3. Clinical correlations: In the limitations section, mention that clinical correlations will be addressed in a future paper and briefly explain why this data was not included in the current study.

4. Pathway analysis: Add details about the tools and parameters used for pathway analysis in the methods section to improve reproducibility.

5. Biomarker validation: Include a paragraph in the discussion outlining future research plans for further validation of the potential biomarkers identified.

6. Statistical analysis: Provide more detailed description of statistical methods in the methods section, particularly regarding multiple comparison corrections.

7. Tryptophan metabolites: Consider expanding the discussion on tryptophan metabolites to include potential clinical implications for PKU patient management.

8. Data availability statement: Add a statement explaining how other researchers can access the raw data from this study, if possible.

These suggestions aim to provide readers with additional context and improve the overall clarity of your manuscript. Please let me know if you have any questions or need further clarification on these points.

Best regards,

Comments on the Quality of English Language

Minor editing of English language required.

Author Response

Comments 1: Sample storage: While you address long-term storage effects in the discussion, consider briefly stating the specific storage duration and conditions in the methods section.

Response 1: Thank you for the suggestion. The year in which each sample was collected is added to Table S5. Furthermore, the storage duration and conditions were already described in the method section pg 3, line 122 and pg 3 line 136. In response to your request, we added that the samples were not thawed during storage (pg 3, line 122) and the year the measurements were conducted, pg 4, line 175 and pg 5, 221-222.

Comments 2: Control group: Please add a paragraph in the methods describing the characteristics of the control group (e.g., age range, gender ratio) to facilitate comparison with the PKU patient cohort.

Response 2: Thank you for your comment. We have already included information about our samples, such as age and gender, in Supplementary Information Table S5. To address your request, we have added specific details about the samples used in the statistical analysis to the Methods section, pg 3-4, lines 140-148.

Comments 3: Clinical correlations: In the limitations section, mention that clinical correlations will be addressed in a future paper and briefly explain why this data was not included in the current study.

Response 3: Thank you for your comment.  In the discussion we described that we aimed to describe the ‘clinical correlations’ in a separate paper. The main reason for not combining this immediately is that this first paper describes our methods in detail and discusses the main PKU biomarkers found, whereas in the second paper we aim to describe the ‘clinical correlation’ in detail, allowing both topics to receive the attention and depth they deserve. Kindly inform us if this does not align with your expectations. The text in the discussion can be found at pg 26, lines 866-872.

Comments 4: Pathway analysis: Add details about the tools and parameters used for pathway analysis in the methods section to improve reproducibility.

Response 4: Thank you for your valuable suggestion. We have previously added a section to the Methods paragraph in the revised manuscript. To accommodate your request, we added more details, e.g. tool and parameter settings, to the section, pg 9, lines 342-353.

Comments 5: Biomarker validation: Include a paragraph in the discussion outlining future research plans for further validation of the potential biomarkers identified.

Response 5: Thank you for your comment. We included a paragraph in the discussion, pg 27, lines 895-907.

Comments 6: Statistical analysis: Provide more detailed description of statistical methods in the methods section, particularly regarding multiple comparison corrections.

Response 6: Thank you for the comment. We tried to clarify the univariate statistical analysis procedure, see pg 6, lines 295-299.

Comments 7: Tryptophan metabolites: Consider expanding the discussion on tryptophan metabolites to include potential clinical implications for PKU patient management.

Response 7: Thank you for the suggestion.

In our discussion, we have deliberately exercised caution with the in-depth interpretation of our findings. The Trp metabolites can be detected in urine, which is less invasive for the patient; this could be advantageous provided that the metabolites prove to be clinically relevant to the neurocognitive functioning of our patients. However, before we engage in an extensive discussion on this topic, we wish to ascertain that these components indeed provide added clinical value.

Comments 8: Data availability statement: Add a statement explaining how other researchers can access the raw data from this study, if possible.

Response 8: Thank you for your comment. For this point, we would like to refer to the Data Availability Statement and the Methods section pg 28, lines 968-970 and pg 5, lines 224-225.